# Phytotoxic Activity of Alkaloids in the Desert Plant *Sophora alopecuroides*

**DOI:** 10.3390/toxins13100706

**Published:** 2021-10-06

**Authors:** Lijing Lei, Yu Zhao, Kai Shi, Ying Liu, Yunxia Hu, Hua Shao

**Affiliations:** 1Chemistry and Environment Science School, Yili Normal University, Yining 835000, China; leilijing5521@hotmail.com; 2Bioscience and Geosciences School, Yili Normal University, Yining 835000, China; satzl2017@163.com (Y.Z.); zylyzhlily@126.com (Y.L.); 3Historical Geography and Tourism School, Shangrao Normal University, Jiangxi 334001, China; 4State Key Laboratory of Desert and Oasis Ecology, Xinjiang Institute of Ecology and Geography, Chinese Academy of Sciences, Urumqi 830011, China; shikai19@mails.ucas.ac.cn; 5University of Chinese Academy of Sciences, Beijing 100049, China; 6Chemistry and Environment Science School, Shangrao Normal University, Jiangxi 334001, China

**Keywords:** allelopathy, alkaloids, indole-3-acetic acid (IAA), cytokinin (CTK), abscisic acid (ABA), malondialdehyde (MDA), antioxidant defense system, *Sophora alopecuroides* L.

## Abstract

*Sophora alopecuroides* is known to produce relatively large amounts of alkaloids; however, their ecological consequences remain unclear. In this study, we evaluated the allelopathic potential of the main alkaloids, including aloperine, matrine, oxymatrine, oxysophocarpine, sophocarpine, sophoridine, as well as their mixture both in distilled H_2_O and in the soil matrix. Our results revealed that all the alkaloids possessed inhibitory activity on four receiver species, i.e., *Amaranthus retroflexus*, *Medicago sativa*, *Lolium perenne* and *Setaria viridis*. The strength of the phytotoxicity of the alkaloids was in the following order: sophocarpine > aloperine > mixture > sophoridine > matrine > oxysophocarpine > oxymatrine (in Petri dish assays), and matrine > mixture > sophocarpine > oxymatrine > oxysophocarpine > sophoridine > aloperine (in pot experiments). In addition, the mixture of the alkaloids was found to significantly increase the IAA content, MDA content and POD activity of *M. sativa* seedlings, whereas CTK content, ABA content, SOD activity and CAT activity of *M. sativa* seedlings decreased markedly. Our results suggest *S. alopecuroides* might produce allelopathic alkaloids to improve its competitiveness and thus facilitate the establishment of its dominance; the potential value of these alkaloids as environmentally friendly herbicides is also discussed.

## 1. Introduction

The genus *Sophora* (*Leguminosae*) comprises more than 70 small tree and shrub species that are widely distributed in the tropical and temperate regions of the two hemispheres [1]. Among them, *Sophora alopecuroides* L., a perennial herb or subshrub, mainly grows in western and central Asia. In China, *S. alopecuroides* (known as Kudouzi) thrives in the arid areas of northwest China as a dominant species [2,3]. The resources of *S. alopecuroides* are relatively rich in Xinjiang province, accounting for 0.61% of the grassland areas [4]. Phytochemical studies of *S. alopecuroides* have led to the isolation of different types of chemical ingredients such as alkaloids, flavonoids, polysaccharides, organic acids, proteins, steroids and so on [5]. Among these compounds, quinolizidine alkaloids were reported to possess the best bioactivity and have been developed into new drugs for a wide range of pharmacological activities, including anti-cancer, anti-inflammatory, anti-fibrosis, anti-virus and anti-arrhythmic activities [1,2].

As with many other medicinal plants, *S. alopecuroides* has been speculated to exhibit allelopathic properties, which presumably facilitates its recent rapid spread in the Yili River Valley of Xinjiang [3,5]. Allelopathy refers to any direct and indirect harmful or beneficial effect by one plant on another through the production of chemical compounds that are released into the nearby environment [6]. It is widely believed that allelopathy may promote the malignant expansion of some invasive plants by releasing allelochemicals such as alkaloids, flavonoids, phenolics, terpenoids, etc., so as to negatively impact neighboring species [7,8,9]. For example, *Oxytropis ochrocephala* Bunge (in the genus *Acanthophora* in the *Leguminous* family) was found to secret allelopathic substances that strongly inhibited growth of *Setaria viridis* as well as other plants, thus resulting in grassland degradation [10]. Previously, extracts of *S. alopecuroides* were found to exhibit inhibitory activity on receiver species [4]. Lv et al. (2012) found that the distilled water extract of *S. alopecuroides* seeds showed a significant suppressive effect on seed germination and seedling growth of *Festuca arundinacea* [11]. Yan et al. (2011) found that both the acetone and distilled water extracts of *S. alopecuroides* inhibited seed germination of *Hippophae rhamnoides* [12]. Studies have shown that *S. alopecuroides* has strong phytotoxic activity, and the production of this activity is closely related to the presence of alkaloids, which is consistent with Wink’s (1983) research that quinolizidine alkaloids are potential inhibitors of seed germination [13]. Quinolizidine alkaloids exist widely in the plant kingdom and are one of the major alkaloids in the genus *Sophora* [14,15]. The alkaloids in *S. alopecuroides* belong to quinolizidine alkaloids, which can be classified into four types, namely matrine-, aloperine-, cytosine-, and lupinine-type alkaloids [1,2]. At present, 43 kinds of alkaloids have been found in *S. alopecuroides*; among them, the vast majority of alkaloids belong to matrine-type alkaloids [16]. It is worth noting that alkaloids such as aloperine (an aloperine-type alkaloid), matrine, oxymatrine, oxysophocarpine, sophocarpine, and sophoridine (matrine-type alkaloids) are relatively abundant in *S. alopecuroides* [17,18,19,20,21] (Figure 1) and are often used for biological activity tests such as pharmacological and antimicrobial activities [22,23,24,25].

*S. alopecuroides* is known to produce relatively large amounts of alkaloids, which are suspected to exhibit allelopathic activity. However, evaluation of the phytotoxicity strength of the alkaloids is still insufficient, as is research into their mechanism. Two dicot plants, *Amaranthus retroflexus* and *Medicago sativa*, as well as two monocot plants, *Lolium perenne* and *Setaria viridis*, were selected as receiver species for this study. They were chosen for the following reasons: firstly, they can be found growing in the same habitats as *S. alopecuroides*; secondly, these four species have previously been used as receiver plants in allelopathic research because of their uniform seedling emergence, high germination rate and their significance in both agricultural and natural fields [3,5,6]. On the other hand, previous work on allelochemical activity in the soil matrix is still insufficient. Based on this, the objectives of our study include: (1) determination of the allelopathic activity of six main alkaloids produced by *S. alopecuroides*; (2) investigation of the alkaloids’ effect on the receiver plant’s hormones and the antioxidant system. Our study will not only help explain the malignant expansion of *S. alopecuroides* in Xinjiang province, but also provide insights into utilizing these alkaloids as environmentally friendly bioherbicides.

## 2. Results

### 2.1. Allelopathic Potential of the Ethanol Extracts of S. alopecuroides and Its Total Alkaloids

The current study focuses on the allelopathic potential of alkaloids produced by *S. alopecuroides*. Due to the fact that seeds are the plant part with the highest content of alkaloids (8.11%), they were chosen for further study [1]. A preliminary experiment was conducted to compare the strength of the allelopathic activity of the ethanol extracts and total alkaloids of *S. alopecuroides* seeds. Starting from 0.3 mg/mL, the ethanol extract started to inhibit seedling growth of *A**. retroflexus* significantly, reducing root elongation by 27.16%. When the concentration reached 3 mg/mL, the length of the roots and shoots of *A. retroflexus* were suppressed by 83.13% and 77.57%, respectively (Figure 2), with an IC_50_ value of 1.40 mg/mL on root growth and 2.04 mg/mL on shoot length. It was evident that the ethanol extract possessed strong phytotoxic activity. We therefore went on to extract the total alkaloids from the ethanol extract to investigate whether they have similar phytotoxic activity.

Total alkaloids also exerted strong inhibitory activity on seedling growth of *A. retroflexus* (Figure 2). The root length of *A. retroflexus* was significantly inhibited by 22.07% when treated with 0.3 mg/mL total alkaloids. When the concentration of total alkaloids increased to 3 mg/mL, the root growth of *A. retroflexus* was reduced by 85.06%, and shoot length was suppressed by 65.23%, with an IC_50_ value of 1.50 mg/mL on root growth and 2.37 mg/mL on shoot length of *A. retroflexus.*

### 2.2. Phytotoxic Effect of Selected Alkaloids via Petri Dish Assay

*Sophora alopecuroides* contains a number of alkaloids, with aloperine, matrine, oxymatrine, oxysophocarpine, sophocarpine, and sophoridine being the most abundant that are frequently reported for their various biological activities such as strong pharmacological activity and antimicrobial activity (Figure 1) [2]. The phytotoxic effect of the above-mentioned six alkaloids along with their mixture (concentrations tested ranged from 20 to 2500 µg/mL) were evaluated against two dicot plants, *A. retroflexus* and *M**. sativa*, as well as two monocot plants, *L**. perenne* and *S**. viridis* (Table 1). For *A. retroflexus*, at 20 µg/mL, aloperine and sophoridine suppressed root growth by 17.30% and 29.37%, respectively. At 100 µg/mL, aloperine, oxymatrine and the mixture significantly reduced root growth of *A. retroflexus* by 46.57%, 23.20% and 59.82%, respectively. When the concentration reached 2500 µg/mL, aloperine, matrine, oxymatrine, oxysophocarpine, sophocarpine, sophoridine and the mixture showed the strongest inhibitory activity that significantly restrained root growth of *A. retroflexus* by 100%, 88.70%, 76.32%, 97.56%, 100%, 85.12% and 88.14%, respectively.

As for *M. sativa*, at 20 µg/mL, sophocarpine promoted root elongation of *M. sativa* by 31.74%, whereas aloperine, matrine, oxymatrine and sophoridine significantly suppressed root growth of *M. sativa* by 42.44%, 26.01%, 31.10% and 28.93%, respectively. When the concentration reached 100 µg/mL, aloperine, matrine, oxymatrine, oxysophocarpine, sophocarpine, sophoridine and the mixture reduced root growth of *M. sativa* by 32.26%, 53.15%, 33.58%, 55.54%, 58.34%, 35.34% and 23.33%, respectively. At the highest concentration tested (2500 µg/mL), aloperine, matrine, oxymatrine, oxysophocarpine, sophocarpine, sophoridine and the mixture showed the strongest inhibitory activity, reducing root growth of *M. sativa* by 98.13%,79.55%, 74.32%, 67.31%, 93.31%, 79.48% and 93.72%, respectively.

Meanwhile, aloperine significantly suppressed root growth of *L. perenne* by 26.54% and 24.87% at 20 and 100 µg/mL, respectively; the effects of other alkaloids on *L. perenne* roots were not significant at the same concentrations tested. When the concentration increased to 500 µg/mL, aloperine, oxysophocarpine and sophocarpine negatively affected root development of *L. perenne* by 52.30%, 14.99% and 50.15%, respectively. At the highest concentration (2500 µg/mL), aloperine, matrine, oxymatrine, oxysophocarpine, sophocarpine, sophoridine and the mixture prohibited root growth of *L. perenne* by 100%, 97.35%, 31.76%, 29.97%, 100%, 72.70% and 97.74%, respectively.

For *S. viridis*, at 20 and 100 µg/mL, the effects of all alkaloids on *S. viridis* roots were not significant. When the concentration reached 500 µg/mL, sophocarpine significantly suppressed root growth of *S. viridis* by 50.15%. At the highest concentration (2500 µg/mL), aloperine, matrine, sophocarpine, sophoridine and the mixture restrained root growth of *S. viridis* by 100%, 89.55%, 94.22%, 85.97% and 91.98%, respectively. In general, shoot development of receiver species showed a similar pattern as root growth, but to a lesser extent.

### 2.3. Phytotoxic Effect of Selected Alkaloids via Pot Experiments

Results obtained from the pot experiments differed significantly from that of the Petri dish assays, most likely due to the effect of the complicated biotic and abiotic processes in soil on the alkaloids (Table 2). For *A. retroflexus*, at 20 µg/g, aloperine, matrine, oxymatrine, oxysophocarpine, sophoridine and the mixture began to suppress root growth of *A. retroflexus* by 13.23%, 19.14%, 33.06%, 15.73%, 15.96% and 34.42%, respectively. At 100 µg/g, oxymatrine, oxysophocarpine, sophoridine and the mixture significantly reduced root growth of *A. retroflexus* by 18.02%, 24.24%, 25.33% and 26.45%, respectively. When the concentration reached 500 µg/g, the effect of the alkaloids was even more potent, with aloperine, oxymatrine, oxysophocarpine, sophoridine and the mixture reducing root elongation of *A. retroflexus* by 14.87%, 52.83%, 26.22%, 25.19% and 16.42%, respectively. At the highest concentration (2500 µg/g), aloperine, matrine, oxymatrine, oxysophocarpine, sophocarpine, sophoridine and the mixture restrained root growth of *A. retroflexus* by 18.10%, 41.34%, 58.92%, 67.04%, 25.00%, 34.90% and 46.00%, respectively.

For *M. sativa*, at 20 µg/g, oxymatrine and sophocarpine significantly inhibited root growth of *M. sativa* by 23.49% and 31.21%, respectively. When the concentration rose to 100 µg/g, oxymatrine, oxysophocarpine and sophocarpine significantly reduced root growth of *M. sativa* by 14.75%, 18.44% and 29.95%, respectively. Meanwhile, oxymatrine, oxysophocarpine, sophocarpine and the mixture significantly suppressed root growth of *M. sativa* by 33.03%, 25.61%, 36.44% and 23.38% at 2500 µg/g, respectively.

*Lolium perenne* seemed to be the most tolerant species when exposed to the alkaloids. In general, only promotive activity was observed when alkaloids were applied: At 20 µg/g, aloperine, matrine, oxymatrine, oxysophocarpine and the mixture promoted root elongation of *L. perenne* by 22.12%, 41.08%, 21.54%, 28.85% and 32.03%, respectively. At 100 µg/g, matrine, oxysophocarpine and sophoridine facilitated root elongation of *L. perenne* by 38.18%, 31.35%, and 24.24%, respectively; when the concentration was increased to 500 µg/g, matrine and sophoridine accelerated root elongation of *L. perenne* by 33.45% and 26.94%, respectively. At the highest concentration (2500 µg/g), the effect of all alkaloids on *L. perenne* root were not significant.

For *S. viridis*, at 20 µg/g, oxymatrine promoted root elongation of *S. viridis* by 24.61%; the effects of other alkaloids on S. viridis root were not significant. At 100 µg/g, although oxymatrine significantly suppressed root growth of *S. viridis* by 23.69%, matrine and the mixture facilitated root elongation of *S. viridis* by 44.48% and 41.65%, respectively. When the concentration reached 500 µg/g, matrine and the mixture stimulated root development of *S. viridis* by 34.09% and 36.07%, respectively. At the highest concentration tested (2500 µg/g), the effect of all alkaloids on *S. viridis* root were not significant.

The results of the Petri dish assays showed that *S. alopecuroides* alkaloids possessed significant phytotoxic activity against receiver plants; however, their strength varied greatly (Table 3). All 7 alkaloids exhibited inhibitory effects against the receiver plants with their strength ranging in the following order: sophocarpine > aloperine > oxysophocarpine > mixture > sophoridine > matrine > oxymatrine, and their corresponding IC_50_ values being 0.692, 0.945, 1.394, 1.451, 1.934, 1.944, and 3.295 mg/mL, respectively. Roots were generally more sensitive to added chemicals compared with shoots; in our work, the effect of the alkaloids on root length of receiver plants were in the following order: sophocarpine > aloperine > mixture > sophoridine > matrine > oxysophocarpine > oxymatrine, with the corresponding IC_50_ values being 0.565, 0.718, 1.069, 1.146, 1.436, 2.168, and 2.340 mg/mL, respectively. In conclusion, alophorine, sophocarpine and mixture exhibited stronger effects than other alkaloids, and the dicot plants (*A. retroflexus* and *M. sativa*) were obviously more sensitive than the monocot plants (*L. perenne* and *S. viridis*). Sophocarpine exerted slightly stronger activity on shoot elongation than root growth, with an IC_50_ value of 0.464 mg/mL on shoot length of *A. retroflexus*, and with an IC_50_ value of 0.155 mg/mL on shoot length of *M. sativa*. Other alkaloids all showed stronger activity on root growth than shoot length.

Compared with the results of the Petri dish assays, the alkaloids in the pot experiments showed relatively weak inhibitory activity (Table 4). The inhibitory effect of selected *S. alopecuroides* alkaloids on the root length of tested plants was ranked as follows: matrine > mixture > sophocarpine > oxymatrine > oxysophocarpine > sophoridine > aloperine, with the corresponding IC_50_ values being 4.626, 4.987, 6.524, 7.476, 8.347, 15.104, and 15.921 mg/mL, respectively. It is also noteworthy to mention that the allelopathic inhibitory effects of matrine, the mixture and sophocarpine were relatively stronger than other alkaloids. The individual alkaloids’ phytotoxicity differed significantly, which is suspected to be related to their chemical structures, and the activity-structure relationship needs more investigation in the future.

Our results revealed that the IC_50_ of the alkaloid mixture was always lower than that of the average of each individual alkaloid. In Petri dish assays, the IC_50_ values of the mixture on roots, shoots, and roots + shoots were slightly smaller than that of the average of each individual alkaloid; however, in pot assays, the IC_50_ values of the mixture were markedly lower than that of the average of each individual alkaloid. For instance, the IC_50_ value of the mixture on root and shoot growth was only half of the IC_50_ of the average of each individual alkaloid, implying that a synergistic effect among these alkaloids might occur, thus enhancing the activity of the mixture.

### 2.4. Phytohormone Content Determination

Changes in the phytohormone content under treatment with mixed alkaloids was determined using *M. sativa* seedlings due to the fact that this species was relatively sensitive, and its biomass was large enough for all the phytohormone content assays (Figure 3). As the concentration of the mixture increased, the content of indole-3-acetic acid (IAA) increased consistently in a dose-dependent manner. At the highest concentration tested (2500 μg/mL), IAA levels were enhanced by 1261.99%. At the same time, the content of cytokinin (CTK) in all the treatments decreased significantly, ranging from 23.56% to 68.86%. In addition, the content of abscisic acid (ABA) diminished gradually along with increase of the alkaloid concentration. At 2500 μg/mL, ABA levels were almost undetectable.

### 2.5. Malondialdehyde (MDA) and Antioxidative Enzyme Activity Determination

The content of MDA and the activity of peroxidase (POD), superoxide dismutase (SOD) and catalase (CAT) responded in a dose-dependent manner to alkaloid treatment (Figure 4). With the increase of alkaloid concentration, the content of MDA and the activity of POD increased gradually. At the highest concentration applied (2500 μg/mL), the content of MDA and the activity of POD was enhanced by 228.4% and 48.09%, respectively. At the same time, the activities of SOD and CAT under alkaloid treatment decreased significantly. When the concentration reached 2500 μg/mL, the activity of SOD and CAT was reduced by 56.13% and 48.36%, respectively.

## 3. Discussion

Many legume plants have been reported to possess allelopathic effects. Nakano et al. (2004) found that leaves of mesquite (*Prosopis juliflora*) contained 3-oxo-juliprosine and 3′-oxo-juliprosine, which significantly inhibited root and shoot growth of cress (*Oenanthe javanica*) seedlings [26]. Adler et al. (2007) found that the growth of corn (*Zea mays*), beans (*Phaseolus vulgaris*), and cabbage (*Brassica oleracea*) were suppressed by the aqueous extracts of fresh velvetbean (*Stizolobium capitatum*) leaves [27]. Hill (2007) reported that cowpea (*Vigna unguiculata*) ethyl acetate extract (8 g/L) reduced germination percentage of common chickweed (*Stellaria media*) and wild carrot (*Daucus carota*) by 32% and 84%, respectively [28]. Our results indicated that alkaloids produced by *S. alopecuroides* possess significant allelopathic potential, not only in an aqueous solution, but also in the soil, implying that alkaloids might be an important group of allelochemicals of this species; the malignant expansion of *S. alopecuroides* in the Yili River Valley of Xinjiang province is possibly induced, at least in part, by allelopathy [1,3].

Legumes are rich in alkaloids, which is one of the main groups of allelochemicals. Wink (1983) found that germination of *Lactuca sativa* was inhibited by over 90% when exposed to 4 mM quinolizidine alkaloids [13]. Villa-Ruano et al. (2012) discovered that both the methanolic and semi-purified alkaloid extracts of *Lupinus jaimehintoniana* had a clear inhibitory effect on *L. sativa* starting from 50 µg/mL, and when it reached 300 µg/mL, seed germination of *L. sativa* was inhibited by 94.4% [29]. Petroski et al. (1990) reported that eight of the synthetic *N*-acyl derivatives of loline were phytotoxic and exhibited a 50% reduction in seed germination and seedling growth of alfalfa (*Medicago sativa*) and annual ryegrass (*Lolium multiflorum*) with doses of less than 1.5 × 10^−7^ mol/seed [30]. Qin et al. (2002) found that oxidized matrine produced by *S. alopecuroides* had an inhibitory effect on barnyard grass (*Echinochloa crusgalli*) seedlings, and sophoridine possessed an inhibitory effect on green bristlegrass (*Setaria viridis*) seedlings [31]. 

In the Petri dish assays, each individual alkaloid and their mixture exhibited potent inhibitory effects on seedling growth of the tested plants in a dose-dependent manner. Among them, sophocarpine is a major ingredient of *S. alopecuroides* with a D ring 13,14-position dehydrogenation analogue of matrine and has a wide range of pharmacological effects [32,33]. The EC_50_ and LC_50_ values of sophocarpine’s teratogenic and lethal effects are 87.1 and 166.0 mg/L on zebrafish embryos/larvae from 0 to 96/120 h post fertilization (hpf) [34]. This suggests that sophocarpine has strong pharmacological activity. The mixture also showed very potent phytotoxic activity, which might be the result of the interaction of multiple mono-alkaloids. This is consistent with Ma’s (2018) research: it was found that both bioassays and field trials proved that the combination of the four monomers from *S. alopecuroides* possesses strikingly aphicidal synergistic action [22]. It is generally believed that individual allelochemicals have a weak inhibitory effect, while mixtures have a synergistic effect [35,36]. Since allelopathy in plants is often the result of the combined action of several compounds, there is often synergistic or antagonistic action between allelopathic substances.

Pot experiments indicated that the alkaloids’ suppressive activity was greatly reduced; in fact, in some occasions, promotive effects were observed, especially on the monocot plants. The soil matrix differed greatly from the aqueous medium: there are numerous soil microorganisms living in the soil which are capable of degrading chemicals including alkaloids. This is consistent with Dou’s (2017) research that matrine has medium adsorption and is easy to move and easy to degrade in soil [37]. We suspect that the decline of the alkaloids’ activity was the consequence of degradation caused by soil microorganisms. In fact, phytotoxins will not function as active allelochemicals unless they can be released into the environment and persist in toxic forms in the medium (soil/air/water) at allelopathic levels for a certain period of time [5]. Therefore, even though it can be demonstrated that these alkaloids are released into the soil matrix, either via leaching, litter decomposition or root exudation, still, like other allelochemicals, the fate of these alkaloids depends greatly on the environment. Once they enter the soil, these chemicals are exposed to various physicochemical and biological processes which might trigger degradations or chemical reactions that lead to the production of novel compounds with different biological activities [38]. 

Phytohormones have a vital role in mediating plant response to abiotic stress, by which the plant may attempt to escape or survive stressful conditions. This may result in reduced growth so that the plant can focus its resources on withstanding the stress [39]. IAA is a main auxin well-known for its essential roles in plant morphogenesis, including tropistic growth, root patterning, vascular tissue differentiation, auxiliary bud formation, and flower organ development [40]. Our study showed that the IAA content of *M. sativa* seedlings increased significantly after alkaloid treatment, which is consistent with Du’s (2013) research showing that the IAA content was decreased after drought stress, but was significantly increased under cold and heat stresses [41]. Bi et al. (2019) reported that the presence of arbuscular mycorrhizal fungi increased IAA and CTK levels in maize (*Zea mays*) roots, reduced ABA levels, and enhanced shoot biomass by 34% [42], which is similar with our results, which showed that selected *S. alopecuroides* alkaloids increased IAA levels and decreased ABA levels in *M. sativa* seedlings. The significant decrease of ABA content observed may be due to the impaired synthesis pathway of the hormone. It is clear that different phytohormones affected overlapping physiological processes, and the physiological effects of phytohormones depended on specific hormone combinations rather than the independent activity of each one [43].

In plants, reactive oxygen species (ROS) are formed as byproducts of many metabolic pathways, especially aerobic energy metabolism, and of plant exposure to various abiotic factors, such as high salinity and extreme cold, heat or drought conditions [44,45]. Under non-stressed conditions, the balance between ROS accumulation and scavenging is maintained by low molecular weight antioxidants and antioxidative enzymes [44]. Some abiotic stress conditions, such as salt stress, can induce oxidative damage, as indicated by higher hydrogen peroxide and MDA contents and electrolyte leakage, by interrupting the antioxidant defense system and promoting the accumulation of toxic levels of Na+ [45]. Lipid peroxidation is a major sign of oxidative damage in plants and can be measured by assessing MDA content [46,47]. Chen et al. (2021) showed that salt stress increased the MDA content by 164% compared with unstressed control plants; however *L*-Glu pretreatment suppressed this salt-stress-induced increase in MDA content by 65% [47]. This is consistent with our results; alkaloids caused yellowish injury on *M. sativa* seedlings, along with an increase in MDA content by 228.4% (Figure 4a), indicating that plant cells were seriously damaged. To conquer the noxious impacts of oxidative stress, plants induced the establishment of antioxidant enzymes; these enzymes function in the guarding system of the plant [48].

Of the antioxidative enzymes, POD plays key roles in cellular ROS detoxification [49]. Gao et al. (2010) found that in the woody plant species *Tamarix hispida*, transcripts of PODs were highly induced by drought in different organs [50]. As is shown in Figure 4c, the activity of POD increased significantly in response to alkaloid treatment, which is consistent with Dui’s (2003) study, which showed that POD activity was maximal at the time of radicle protrusion and seedling development [51]. In another study, Li et al. (2011) also found that mild drought preconditioning increased POD activity and MDA content [52]. Glycine betaine (GB) stabilizes the quaternary structure of proteins and enzymes, improves the antioxidant defense system, photosystem-II, and reduces membrane permeability and H_2_O_2_-mediated signaling. Islam et al. (2021) showed that application of 20 mM GB significantly improved the activity of CAT by 10.06% and of SOD by 16.89% over the control [53]. In this study, with an increase in alkaloid concentration, the activities of SOD and CAT gradually decreased (Figure 4b,d), indicating that the antioxidant mechanism of cells was damaged and excessive ROS could not be effectively removed. These findings are similar to Ji’s (2011) study, which showed that after 40 h of treatment with *Lactarius vellereus* fermentation liquid extract, the activities of SOD and CAT of *Alternaria alternata* decreased rapidly to the minimum (0.828 U∙g^−1^, 0.012 U∙g^−1^∙s^−1^, respectively) [54]. Our results were also similar with other reports: Ibrahim et al. (2018) discovered that high salinity significantly reduced SOD activity of wheat (*Triticum aestivum*) [55]; Yan et al. (2006) found that CAT activity of liquorice (*Glycyrrhiza uralensis*) decreased under salt and drought stress [56]; and Fang et al. (2018) found that metalaxyl significantly inhibited root activity and significantly improved leaf SOD, POD, and CAT activities and MDA content in tobacco (*Nicotiana tabacum*) seedlings [57]. Apparently, application of the alkaloids triggered a series of physiological shifts in the seedlings to prepare for the stress, which was expressed in a dose-dependent manner.

## 4. Conclusions

Alkaloids produced by *S. alopecuroides* were found to possess allelopathic effects against receiver plants; however, the strength of the alkaloids when applied in the aqueous medium was much stronger than in the soil matrix. Sophocarpine and the mixture of the alkaloids were found to possess more potent inhibitory activity compared with other alkaloids in both mediums; the mixture of alkaloids was also found to exert a phytotoxic effect via altering the content of IAA, CTK and MDA, as well as the activity of POD, SOD and CAT. Our results implied that alkaloids produced by *S. alopecuroides* have the potential to be further explored as environmentally friendly herbicides. Our work also suggests that *S. alopecuroides* might produce allelopathic alkaloids to improve its competitiveness and thus facilitate the establishment of its dominance.

## 5. Materials and Methods

### 5.1. Material and Reagents

Whole plants of *S. alopecuroides* with mature fruits were collected in suburban Changji city, Xinjiang province, China, in July 2020 (Lat 43.9315 N, Lon 87.3483 E, with an elevation of 602.67 m). Plants were identified by Prof. Li Wenjun from the Xinjiang Institute of Ecology and Geography, Chinese Academy of Sciences. Plant materials were air dried in our laboratory in the shade at room temperature for two weeks before use. Aloperine, matrine, oxymatrine, oxysophocarpine, sophocarpine, and sophoridine (98% purity) were purchased from Desite Biology Co., Ltd. (Chengdu, Sichuan, China).

### 5.2. Preparation of the Ethanol Extract and the Total Alkaloids 

Fifty g of *S. alopecuroides* seeds were harvested from the plant materials and ground into powder, which was then extracted with 95% ethanol under sonication for 30 min, 3 times at room temperature. The ethanol extract was concentrated under reduced pressure to yield 5.6 g of a dark brown residue which was subsequently dissolved in HCl (5%, 250 mL) and filtered. The filtrate was partitioned 3 times with chloroform (600 mL). The aqueous acid layer was made alkaline to pH = 9 with NH_4_OH (25%), then extracted 4 times with chloroform (1000 mL) to yield 0.8 g of chloroform extract. The chloroform extract was then recrystallized in ethanol to give 24 mg of yellowish total alkaloids. The ethanol extract and the total alkaloids were then tested for their phytotoxicity against *A. retroflexus.* They were diluted in methanol to give 0, 0.3 and 3 mg/mL concentrations. Test seeds were first surface sterilized using 75% ethanol before use. Five mL corresponding methanol diluent were then added to Petri dishes (9 cm in diameter) lined with Whatman No. 1 filter papers. After complete evaporation of methanol, distilled water (5 mL) was added to each Petri dish followed by the addition of 10 seeds. Petri dishes were all sealed with Parafilm to prevent water loss and stored in the dark at 25 °C. Seedlings were allowed to grow for 7 days before shoot and root lengths were measured. Three replicates were made for all bioassays. In total, 30 seedlings were measured.

### 5.3. Phytotoxic Effect of Each Individual Alkaloid and Their Mixture 

Aloperine, matrine, oxymatrine, oxysophocarpine, sophocarpine, sophoridine and their mixture (1:1:1:1:1:1:1) were diluted in methanol to give 0, 20, 100, 500 and 2500 μg/mL concentrations. Their phytotoxic activity was measured against two dicot plants, *A. retroflexus* and *M. sativa*, and two monocot plants, *L. perenne* and *S. viridis*, using the same above-mentioned bioassay procedure. In the pot experiments, alkaloids were tested at the same concentrations, which were first homogenized in distilled H_2_O and then mixed thoroughly with the soil. In detail, each pot (10 cm in diameter) was filled with 30 g of sterilized soil and then 10 test seeds of test species were sown. Seedlings were allowed to grow for 2 weeks before measurement in the greenhouse under natural conditions. Three replicates were made for all bioassays. In total, 30 seedlings were measured.

### 5.4. Phytohormone Content Determination

Fresh *M. sative* seedlings were used to determine phytohormone content. Seedlings were weighed (0.5 g) and ground, and 2 mL of pre-cooled 80% methanol was added to the homogenate. After sealing with plastic wrap, the sample was cold soaked overnight at 4 °C, and then centrifuged at 5000 r/min at 4 °C for 10 min. The supernatant was extracted with 80% methanol, and the supernatant was combined by vibration and ultrasound twice. 4 °C nitrogen was blown into the aqueous phase. The aqueous phase was subsequently extracted with ethyl acetate 3 times. Then, the aqueous phase was combined with ethyl acetate. The aqueous phase was blown dry with nitrogen at 4 °C, and the acetic acid solution with pH = 3.5 was added. The sample was determined later. 

### 5.5. Malondialdehyde (MDA) and Antioxidative Enzyme Activity Determination

Oxidative damage to cell membrane lipids was evaluated by determining the content of malondialdehyde (MDA). MDA activity was performed according to the MDA reagent kit (YX-W-A401, Sino Best Biological Technology Co. Ttd, Shanghai, China). In detail, 3.0 g frozen plant sample was crashed in liquid nitrogen, extracted in 5 mL of 10% trichloroacetic acid and centrifuged at 10.000× *g* for 25 min at 4 °C. Then, 1 mL supernatant was added into 2 mL of 0.67% 2-thio-barbituric acid (prepared in 0.05 mM NaOH) and mixed. The mixture was incubated in boiled water bath for 15 min and then cooled to room temperature. The absorbance was measured at 450 nm and 532 nm against a reagent blank at 25 °C. The MDA content in the plant was expressed as nmol/g fresh weight.

Superoxide dismutase (SOD) activity was performed according to the SOD reagent kit (YX-W-A500, Sino Best Biological Technology Co. Ttd, Shanghai, China). Frozen plant samples were weighed (3.0 g) and crushed in liquid nitrogen containing 1% polyvinyl polypyrrolidone (PVPP); then, 0.1 g grinded sample was transferred to 1 mL of extract buffer and vortexed for 3 min, followed by centrifugation at 8000× *g* for 10 min at 4 °C. The supernatant was measured by recording the absorbance at 560 nm at 25 °C. The SOD activity was expressed as U/g fresh weight.

Peroxidase (POD) activity was assayed according to the POD reagent kit (YX-W-A502, Sino Best Biological Technology Co. Ttd, Shanghai, China) method. Samples were prepared as per the SOD protocol. The supernatant was then measured by recording the increase in absorbance at 420 nm at 25 °C. The POD activity was expressed as U/g fresh weight.

Catalase (CAT) activity was assayed according to the CAT reagent kit (YX-W-A501, Sino Best Biological Technology Co. Ttd, Shanghai, China). In detail, 3.0 g of frozen plant sample was crushed in liquid nitrogen containing 1% PVPP. Then, 0.1 g grinded sample was transferred to 1 mL of extract buffer and incubated in a water bath of 37 °C for 3–5 min. Then 0.1 mL of 20 mmol L^−1^ H_2_O_2_ in 50 mmol L^−1^ phosphate butter (pH 7.5) was added into the above reaction solution and vortexed, followed by centrifugation at 8000 × g for 10 min at 4 °C. The supernatant was measured by recording the decrease in H_2_O_2_ in absorbance at 405 nm at 25 °C. The CAT activity was expressed as U/g fresh weight.

### 5.6. Statistical Analyses

The bioassay results were expressed as mean ± standard error (SE) of the mean. One-way ANOVA (*p* < 0.05) was applied using the SPSS statistical package version 13.0 (SPSS Inc., Chicago, IL, USA) for Windows to examine whether the difference of the phytotoxic effects of aloperine, matrine, oxymatrine, oxysophocarpine, sophocarpine, sophoridine and their mixture on seedling growth of test species at different concentrations was significant; then, all the above-mentioned data was further processed using Fisher’s LSD test at *p* < 0.05 level to compare the difference among treatments. The inhibitory concentration required for 50% inhibition (IC_50_) values was calculated using probit analysis (SAS Institute. SAS/STAT User’s Guide).

## Figures and Tables

**Figure 1 toxins-13-00706-f001:**
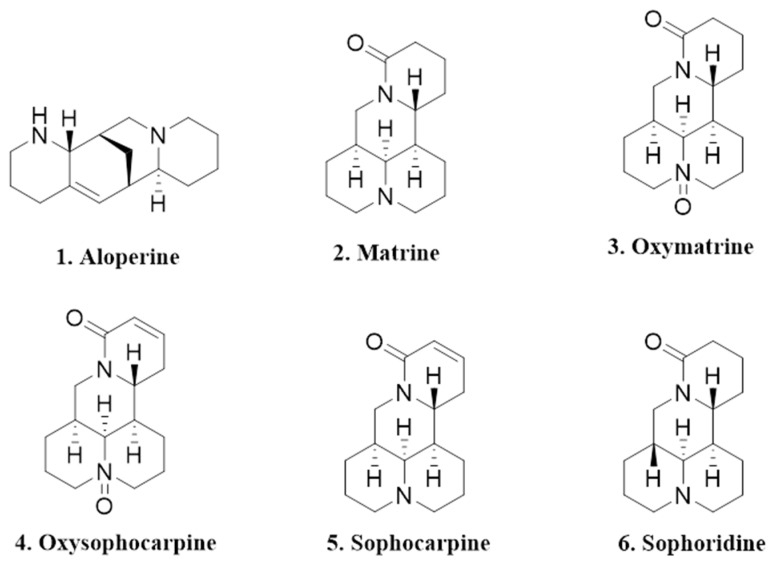
Chemical structures of the selected six alkaloids.

**Figure 2 toxins-13-00706-f002:**
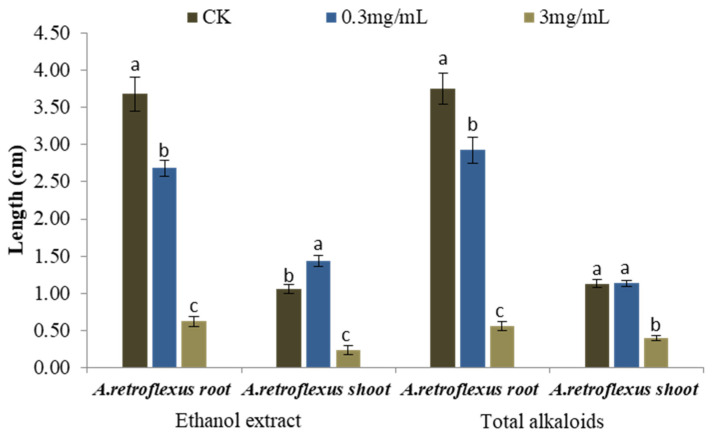
Phytotoxic activity of the ethanol extracts and total alkaloids from seeds of *S. alopecuroides* tested at 0 mg/mL (CK), 0.3 mg/mL, 3 mg/mL on root and shoot elongation of *A. retroflexus*. Each value is the mean of three replicates ± SE (n = 30). Means with different letters (a, b, c) indicate significant differences (*p* < 0.05) according to Fisher’s LSD test.

**Figure 3 toxins-13-00706-f003:**
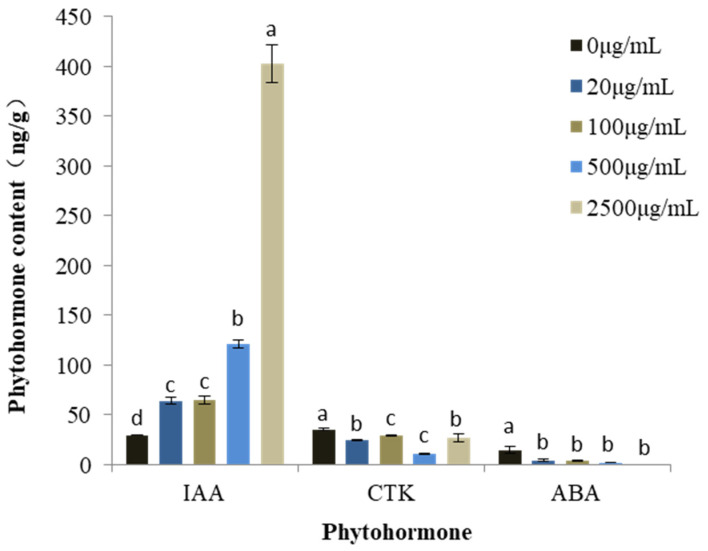
Changes in phytohormone content in *M. sativa* seedlings treated with mixed alkaloids at different concentrations. Each value is the mean of three replicates ± SE (n = 30). Means with different letters (a, b, c, etc.) indicate significant differences at *p* < 0.05 level according to Fisher’s LSD test.

**Figure 4 toxins-13-00706-f004:**
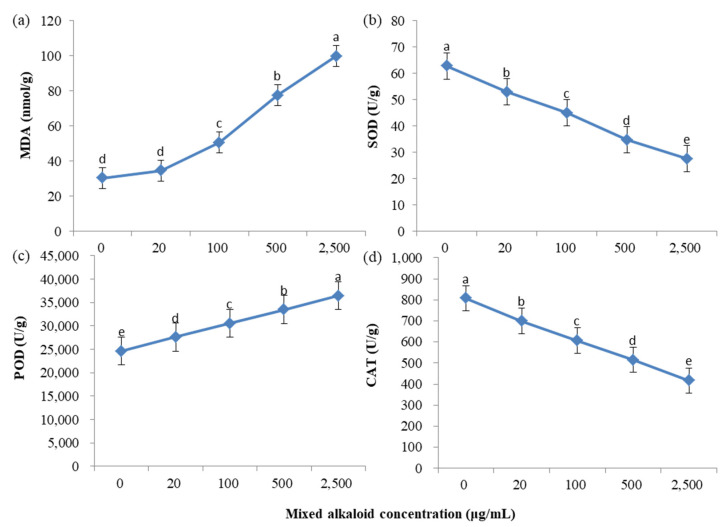
Changes in malondialdehyde (MDA) content and antioxidative enzyme activity in *M. sativa* seedlings treated with mixed alkaloids at different concentrations. (**a**) Change in MDA concent in *M. sativa* seedlings treated with mixed alkaloids at different concentrations. (**b**) Change in SOD activity in *M. sativa* seedlings treated with mixed alkaloids at different concentrations. (**c**) Change in POD activity in *M. sativa* seedlings treated with mixed alkaloids at different concentrations. (**d**) Change in CAT activity in *M. sativa* seedlings treated with mixed alkaloids at different concentrations. Each value is the mean of three replicates ± SE. Means with different letters (a, b, c, etc.) indicate significant differences at *p* < 0.05 level according to Fisher’s LSD test.

**Table 1 toxins-13-00706-t001:** Phytotoxic effects of selected *S. alopecuroides* alkaloids on receiver species via Petri dish assay.

Alkaloids	Concentration	(µg/mL)	*Lolium perenne*		*Amaranthus retroflexus*		*Medicago sativa*		*Setaria viridis*	
			Root	Shoot	Root	Shoot	Root	Shoot	Root	Shoot
**Aloperine**	0		3.69 ± 0.32 ^a^	3.57 ± 0.21 ^a^	3.19 ± 0.13 ^a^	2.87 ± 0.10 ^a^	2.81 ± 0.19 ^a^	2.47 ± 0.12 ^a^	1.63 ± 0.14 ^a^	6.11 ± 0.42 ^a^
	20		2.71 ± 0.41 ^b^	2.77 ± 0.22 ^b^	2.64 ± 0.30 ^b^	2.34 ± 0.25 ^b^	1.62 ± 0.32 ^b^	2.54 ± 0.13 ^a^	1.48 ± 0.33 ^a^	6.31 ± 0.44 ^a^
	100		2.77 ± 0.22 ^b^	2.73 ± 0.28 ^a^	1.71 ± 0.23 ^c^	2.11 ± 0.29 ^b^	1.90 ± 0.10 ^b^	1.53 ± 0.28 ^b^	1.29 ± 0.23 ^a^	4.95 ± 0.39 ^a^
	500		1.76 ± 0.37 ^c^	1.80 ± 0.35 ^b^	0.14 ± 0.05 ^d^	0.29 ± 0.10 ^c^	0.63 ± 0.09 ^c^	1.03 ± 0.16 ^c^	1.04 ± 0.12 ^a^	4.92 ± 0.37 ^a^
	**2500**		** 0.00 ± 0.00 ^d^**	** 0.63 ± 0.18 ^c^**	** 0.00 ± 0.00 ^d^**	** 0.00 ± 0.00 ^c^**	** 0.05 ± 0.03 ^d^**	** 0.07 ± 0.03 ^d^**	** 0.00 ± 0.00 ^b^**	** 1.13 ± 0.47 ^b^**
Matrine	0		4.49 ± 0.48 ^a^	3.17 ± 0.28 ^a^	2.43 ± 0.17 ^a^	2.14 ± 0.10 ^a^	2.81 ± 0.19 ^a^	2.47 ± 0.12 ^a^	2.34 ± 0.75 ^a^	4.59 ± 0.58 ^b^
	20		4.45 ± 0.26 ^a^	3.41 ± 0.17 ^a^	2.34 ± 0.10 ^a^	1.86 ± 0.07 ^a,b^	2.08 ± 0.28 ^b^	2.66 ± 0.22 ^a^	2.18 ± 0.45 ^a^	7.12 ± 0.68 ^a^
	100		4.17 ± 0.46 ^a^	3.12 ± 0.35 ^a^	2.12 ± 0.22 ^a^	2.13 ± 0.13 ^a^	1.31 ± 0.16 ^c^	2.18 ± 0.17 ^a^	2.00 ± 0.31 ^a^	5.39 ± 0.70 ^a,b^
	500		3.44 ± 0.43 ^a^	3.70 ± 0.36 ^a^	1.47 ± 0.10 ^b^	1.80 ± 0.11 ^b^	1.72 ± 0.10 ^b,c^	2.67 ± 0.21 ^a^	2.51 ± 0.57 ^a^	5.17 ± 0.52 ^a,b^
	2500		0.12 ± 0.04 ^b^	2.96 ± 0.29 ^a^	0.28 ± 0.04 ^c^	0.27 ± 0.08 ^c^	0.57 ± 0.03 ^d^	0.45 ± 0.14 ^b^	0.24 ± 0.06 ^b^	3.71 ± 1.01 ^b^
Oxymatrine	0		4.49 ± 0.48 ^a^	3.17 ± 0.28 ^a^	2.43 ± 0.17 ^a^	2.14 ± 0.10 ^a^	2.81 ± 0.19 ^a^	2.47 ± 0.12 ^a^	1.66 ± 0.21 ^a^	5.15 ± 0.66 ^b^
	20		4.41 ± 0.30 ^a^	3.67 ± 0.15 ^a^	2.09 ± 0.14 ^a,b^	1.97 ± 0.11 ^a^	1.93 ± 0.25 ^b^	2.29 ± 0.18 ^a^	2.37 ± 0.28 ^a^	6.48 ± 0.5 ^a,b^
	100		3.63 ± 0.36 ^a,b^	3.56 ± 0.29 ^a^	1.87 ± 0.14 ^b^	2.01 ± 0.10 ^a^	1.86 ± 0.13 ^b^	2.70 ± 0.11 ^a^	2.19 ± 0.27 ^a^	7.08 ± 0.45 ^a^
	500		3.40 ± 0.34 ^a,b^	3.41 ± 0.28 ^a^	1.26 ± 0.10 ^c^	1.53 ± 0.13 ^b^	0.75 ± 0.18 ^c^	1.36 ± 0.32 ^b^	2.03 ± 0.69 ^a^	5.50 ± 0.32 ^a,b^
	2500		3.06 ± 0.33 ^b^	2.93 ± 0.27 ^a^	0.58 ± 0.09 ^d^	1.17 ± 0.17 ^c^	0.72 ± 0.05 ^c^	0.79 ± 0.14 ^c^	1.81 ± 0.22 ^a^	5.73 ± 0.55 ^a,b^
Oxysophocarpine	0		3.69 ± 0.32 ^a^	3.57 ± 0.21 ^a^	3.19 ± 0.13 ^a^	2.87 ± 0.10 ^a^	2.81 ± 0.10 ^a^	2.47 ± 0.14 ^a^	1.63 ± 0.14 ^a^	6.11 ± 0.42 ^a^
	20		3.32 ± 0.31 ^a^	2.99 ± 0.30 ^a^	2.08 ± 0.14 ^a^	2.04 ± 0.17 ^a,b^	2.39 ± 0.05 ^c^	2.41 ± 0.22 ^b^	1.76 ± 0.24 ^a^	6.33 ± 0.35 ^a^
	100		3.36 ± 0.24 ^a,b,c^	3.41 ± 0.23 ^a^	2.01 ± 0.18 ^a^	2.30 ± 0.15 ^a^	1.25 ± 0.07 ^c^	2.21 ± 0.11 ^b^	1.22 ± 0.15 ^a^	6.47 ± 0.44 ^a^
	500		3.14 ± 0.25 ^b,c^	3.09 ± 0.19 ^a^	1.12 ± 0.12 ^b^	1.54 ± 0.13 ^b,c^	1.50 ± 0.00 ^d^	2.22 ± 0.00 ^c^	1.71 ± 0.22 ^a^	5.36 ± 0.84 ^a^
	2500		2.59 ± 0.22 ^c^	3.33 ± 0.26 ^a^	0.08 ± 0.07 ^c^	0.18 ± 0.09 ^c^	0.92 ± 0.07 ^b^	2.50 ± 0.17 ^a^	1.21 ± 0.24 ^a^	5.11 ± 0.40 ^a^
**Sophocarpine**	0		2.16 ± 0.40 ^a^	1.87 ± 0.36 ^a^	3.13 ± 0.20 ^a,b^	2.59 ± 0.15 ^a^	1.02 ± 0.12 ^b^	2.92 ± 0.24 ^a^	2.09 ± 0.22 ^a^	5.50 ± 0.45 ^a^
	20		2.43 ± 0.46 ^a^	2.22 ± 0.37 ^a^	3.55 ± 0.13 ^a^	2.73 ± 0.09 ^a^	1.34 ± 0.09 ^a^	3.16 ± 0.19 ^a^	1.96 ± 0.28 ^a,b^	5.33 ± 0.44 ^a^
	100		1.74 ± 0.40 ^a,b^	1.82 ± 0.51 ^a^	2.73 ± 0.18 ^b^	2.46 ± 0.22 ^a^	0.42 ± 0.08 ^c^	0.37 ± 0.09 ^b^	1.58 ± 0.22 ^a,b^	5.05 ± 0.42 ^a^
	500		1.08 ± 0.33 ^b^	1.61 ± 0.55 ^a^	1.53 ± 0.14 ^c^	1.19 ± 0.10 ^b^	0.42 ± 0.06 ^c^	0.40 ± 0.05 ^b^	1.44 ± 0.20 ^b^	4.56 ± 0.45 ^a^
	**2500**		** 0.00 ± 0.00 ^c^**	** 0.00 ± 0.00 ^b^**	** 0.00 ± 0.00 ^d^**	** 0.00 ± 0.00 ^c^**	** 0.07 ± 0.03 ^d^**	** 0.08 ± 0.04 ^b^**	** 0.12 ± 0.04 ^c^**	** 0.24 ± 0.07 ^b^**
Sophoridine	0		4.49 ± 0.48 ^a^	3.17 ± 0.28 ^a^	2.43 ± 0.17 ^a^	2.14 ± 0.10 ^a^	2.81 ± 0.19 ^a^	2.47 ± 0.12 ^a^	1.66 ± 0.21 ^a^	5.15 ± 0.66 ^a,b^
	20		4.63 ± 0.42 ^a^	3.15 ± 0.29 ^a^	1.72 ± 0.22 ^b^	1.86 ± 0.15 ^a^	1.99 ± 0.17 ^b^	2.50 ± 0.14 ^a^	1.91 ± 0.15 ^a^	7.00 ± 0.52 ^a^
	100		4.45 ± 0.27 ^a^	3.69 ± 0.30 ^a^	1.97 ± 0.17 ^a,b^	1.80 ± 0.09 ^a^	1.81 ± 0.12 ^b^	2.79 ± 0.12 ^a^	1.59 ± 0.39 ^a^	6.47 ± 0.53 ^a^
	500		3.83 ± 0.38 ^a^	3.59 ± 0.24 ^a^	1.06 ± 0.13 ^c^	1.39 ± 0.15 ^b^	1.70 ± 0.11 ^b^	2.88 ± 0.14 ^a^	1.10 ± 0.42 ^a^	4.88 ± 0.69 ^a,b^
	2500		1.23 ± 0.23 ^b^	3.62 ± 0.36 ^a^	0.36 ± 0.09 ^d^	0.58 ± 0.08 ^c^	0.58 ± 0.07 ^c^	0.69 ± 0.18 ^b^	0.23 ± 0.09 ^b^	3.57 ± 1.03 ^b^
**Mixture**	0		4.49 ± 0.48 ^a^	3.17 ± 0.28 ^b^	2.43 ± 0.17 ^a^	2.14 ± 0.10 ^a^	2.81 ± 0.19 ^a^	2.47 ± 0.12 ^a^	1.66 ± 0.21 ^a^	5.15 ± 0.66 ^b^
	20		4.08 ± 0.37 ^a^	4.01 ± 0.23 ^a^	2.00 ± 0.23 ^a^	1.86 ± 0.10 ^a,b^	2.43 ± 0.24 ^a,b^	2.55 ± 0.25 ^a^	2.35 ± 0.33 ^a^	6.10 ± 0.41 ^a,b^
	100		4.25 ± 0.39 ^a^	4.29 ± 0.26 ^a^	0.98 ± 0.15 ^b^	1.53 ± 0.17 ^b,c^	2.15 ± 0.12 ^b^	2.54 ± 0.10 ^a^	2.05 ± 0.34 ^a^	6.44 ± 0.56 ^a,b^
	500		4.76 ± 0.39 ^a^	3.60 ± 0.22 ^a,b^	0.58 ± 0.12 ^b,c^	1.33 ± 0.22 ^c^	0.62 ± 0.13 ^c^	1.09 ± 0.25 ^b^	1.93 ± 0.24 ^a^	6.99 ± 0.39 ^a^
	**2500**		** 0.10 ± 0.04 ^b^**	2.29 ± 0.33 ^c^	** 0.29 ± 0.01 ^c^**	** 0.35 ± 0.03 ^d^**	** 0.18 ± 0.04 ^c^**	** 0.17 ± 0.05 ^c^**	** 0.13 ± 0.06 ^b^**	3.43 ± 0.43 ^c^

Each value is the mean of three replicates ± SE (*n* = 30). Means with different letters (a, b, c, etc.) indicate significant differences at *p* < 0.05 level according to Fisher’s LSD test. Bolded values represent the most potent effects of the alkaloids with the most significant phytotoxic activity.

**Table 2 toxins-13-00706-t002:** Allelopathic effects of selected *S. alopecuroides* alkaloids on receiver species via pot experiments.

Alkaloids	Concentration (µg/g)	*Lolium perenne*		*Amaranthus retroflexus*		*Medicago sativa*		*Setaria viridis*	
		Root	Shoot	Root	Shoot	Root	Shoot	Root	Shoot
Aloperine	0	2.86 ± 0.14 ^b^	6.02 ± 0.33 ^b^	1.94 ± 0.10 ^a^	4.23 ± 0.14 ^a^	2.21 ± 0.11 ^a^	4.31 ± 0.09 ^c^	1.69 ± 0.10 ^a^	8.95 ± 0.26 ^a,b^
	20	3.49 ± 0.17 ^a^	7.24 ± 0.25 ^a^	1.68 ± 0.07 ^b^	4.28 ± 0.17 ^a^	1.95 ± 0.14 ^a^	5.09 ± 0.14 ^a,b^	2.10 ± 0.20 ^a^	9.08 ± 0.27 ^a,b^
	100	3.27 ± 0.17 ^a b^	7.36 ± 0.25 ^a^	1.74 ± 0.07 ^a,b^	3.98 ± 0.16 ^a,b^	1.97 ± 0.11 ^a^	4.76 ± 0.14 ^b^	1.67 ± 0.22 ^a^	9.24 ± 0.34 ^a^
	500	2.91 ± 0.19 ^b^	7.33 ± 0.25 ^a^	1.65 ± 0.09 ^b^	3.94 ± 0.21 ^a,b^	2.17 ± 0.10 ^a^	5.27 ± 0.14 ^a^	1.83 ± 0.15 ^a^	8.78 ± 0.36 ^a,b^
	2500	2.97 ± 0.13 ^b^	6.87 ± 0.25 ^a^	1.59 ± 0.08 ^b^	3.71 ± 0.12 ^b^	2.00 ± 0.10 ^a^	4.73 ± 0.18 ^b^	1.81 ± 0.12 ^a^	8.08 ± 0.27 ^b^
**Matrine**	0	2.86 ± 0.14 ^b^	6.02 ± 0.33 ^c^	1.94 ± 0.10 ^a^	4.23 ± 0.14 ^a^	2.21 ± 0.11 ^a,b^	4.31 ± 0.09 ^c^	1.69 ± 0.10 ^b^	8.95 ± 0.26 ^a^
	20	4.03 ± 0.22 ^a^	8.67 ± 0.29 ^a^	1.57 ± 0.09 ^b^	4.11 ± 0.19 ^a^	2.07 ± 0.13 ^b^	4.67 ± 0.12 ^a,b^	2.10 ± 0.20 ^a,b^	8.30 ± 0.43 ^a,b^
	100	3.95 ± 0.19 ^a^	8.30 ± 0.38 ^a,b^	2.02 ± 0.13 ^a^	4.11 ± 0.22 ^a^	2.27 ± 0.09 ^a,b^	4.96 ± 0.10 ^a^	2.44 ± 0.15 ^a^	8.38 ± 0.22 ^a,b^
	500	3.81 ± 0.19 ^a^	7.86 ± 0.37 ^a,b^	1.73 ± 0.17 ^a,b^	4.25 ± 0.15 ^a^	2.50 ± 0.15 ^a^	4.57 ± 0.12 ^b,c^	2.26 ± 0.15 ^a^	7.80 ± 0.22 ^b,c^
	**2500**	3.27 ± 0.21 ^b^	7.52 ± 0.24 ^b^	** 1.14 ± 0.11 ^c^**	3.51 ± 0.23 ^b^	2.17 ± 0.10 ^a,b^	4.39 ± 0.11 ^b,c^	1.73 ± 0.11 ^b^	7.33 ± 0.49 ^c^
Oxymatrine	0	2.86 ± 0.14 ^b^	6.02 ± 0.33 ^b^	1.94 ± 0.10 ^a^	4.23 ± 0.14 ^a^	2.21 ± 0.11 ^a^	4.31 ± 0.09 ^a^	1.69 ± 0.10 ^b^	8.95 ± 0.26 ^a^
	20	3.47 ± 0.16 ^a^	7.41 ± 0.20 ^a^	1.30 ± 0.06 ^b^	3.29 ± 0.13 ^b^	1.69 ± 0.12 ^b,c^	3.69 ± 0.12 ^b^	2.10 ± 0.07 ^a^	7.83 ± 0.17 ^b^
	100	2.86 ± 0.19 ^b^	8.09 ± 0.21 ^a^	1.59 ± 0.10 ^b^	3.62 ± 0.15 ^b^	1.88 ± 0.10 ^b,c^	4.02 ± 0.10 ^a^	1.29 ± 0.09 ^c^	6.54 ± 0.31 ^c^
	500	2.90 ± 0.18 ^b^	7.96 ± 0.24 ^a^	0.91 ± 0.08 ^c^	3.46 ± 0.25 ^b^	1.72 ± 0.09 ^b^	3.62 ± 0.11 ^b^	1.62 ± 0.10 ^b^	7.29 ± 0.31 ^b^
	2500	2.81 ± 0.14 ^b^	7.59 ± 0.26 ^a^	0.80 ± 0.08 ^c^	1.94 ± 0.32 ^c^	1.48 ± 0.08 ^c^	3.28 ± 0.11 ^c^	1.54 ± 0.08 ^b^	5.80 ± 0.29 ^c^
Oxysophocarpine	0	2.86 ± 0.14 ^b^	6.02 ± 0.33 ^b^	1.94 ± 0.10 ^a^	4.23 ± 0.14 ^a^	2.21 ± 0.11 ^a^	4.31 ± 0.09 ^a,b,c^	1.69 ± 0.10 ^a^	8.95 ± 0.26 ^a^
	20	3.68 ± 0.17 ^a^	8.28 ± 0.23 ^a^	1.63 ± 0.10 ^b^	3.66 ± 0.18 ^b^	2.07 ± 0.13 ^a,b^	4.61 ± 0.11 ^a^	1.74 ± 0.12 ^a^	8.33 ± 0.29 ^a b^
	100	3.75 ± 0.17 ^a^	8.35 ± 0.34 ^a^	1.47 ± 0.08 ^b^	3.96 ± 0.16 ^a,b^	1.80 ± 0.10 ^b,c^	4.53 ± 0.14 ^a,b^	1.78 ± 0.13 ^a^	7.88 ± 0.39 ^b^
	500	3.11 ± 0.18 ^b^	8.20 ± 0.32 ^a^	1.43 ± 0.10 ^b^	3.83 ± 0.18 ^a,b^	1.96 ± 0.14 ^a,b,c^	4.21 ± 0.14 ^b,c^	1.62 ± 0.11 ^a^	7.68 ± 0.26 ^b^
	2500	3.17 ± 0.19 ^b^	8.49 ± 0.26 ^a^	0.64 ± 0.07 ^c^	2.77 ± 0.26 ^c^	1.64 ± 0.09 ^c^	4.01 ± 0.10 ^c^	1.60 ± 0.12 ^a^	7.85 ± 0.32 ^b^
**Sophocarpine**	0	4.62 ± 0.24 ^a,b^	7.14 ± 0.33 ^a,b^	1.64 ± 0.09 ^a^	3.93 ± 0.21 ^b^	2.53 ± 0.19 ^a^	4.46 ± 0.20 ^a^	2.09 ± 0.16 ^a^	6.74 ± 0.36 ^c^
	20	4.67 ± 0.46 ^a,b^	8.31 ± 0.21 ^a^	1.75 ± 0.07 ^a^	4.24 ± 0.12 ^a,b^	1.74 ± 0.08 ^b^	4.59 ± 0.15 ^a^	2.03 ± 0.10 ^a^	9.1 ± 0.27 ^a^
	100	4.44 ± 0.24 ^a,b^	7.99 ± 0.70 ^a^	1.53 ± 0.06 ^a^	4.47 ± 0.17 ^a^	1.77 ± 0.11 ^b^	4.65 ± 0.22 ^a^	1.84 ± 0.20 ^a^	8.33 ± 0.36 ^a^
	500	4.97 ± 0.12 ^a^	7.18 ± 0.41 ^a,b^	1.63 ± 0.06 ^a^	4.38 ± 0.20 ^a,b^	1.49 ± 0.08 ^b^	4.88 ± 0.16 ^a^	2.03 ± 0.11 ^a^	8.07 ± 0.27 ^a,b^
	**2500**	3.95 ± 0.51 ^b^	6.54 ± 0.51 ^b^	**1.23 ± 0.07 ^b^**	3.44 ± 0.12 ^c^	** 1.61 ± 0.07 ^b^**	4.52 ± 0.11 ^a^	2.16 ± 0.12 ^a^	7.18 ± 0.58 ^b,c^
Sophoridine	0	2.86 ± 0.14 ^b^	6.02 ± 0.33 ^c^	1.94 ± 0.10 ^a^	4.23 ± 0.14 ^a^	2.21 ± 0.11 ^a^	4.31 ± 0.09 ^c^	1.69 ± 0.10 ^a^	8.95 ± 0.26 ^a^
	20	3.11 ± 0.19 ^a,b^	8.49 ± 0.35 ^a^	1.63 ± 0.08 ^b^	3.82 ± 0.11 ^b^	2.27 ± 0.13 ^a^	4.93 ± 0.11 ^a^	2.14 ± 0.20 ^a^	8.82 ± 0.38 ^a^
	100	3.55 ± 0.19 ^a^	8.56 ± 0.34 ^a^	1.45 ± 0.10 ^b,c^	3.32 ± 0.23 ^c^	2.54 ± 0.12 ^a^	4.58 ± 0.14 ^b,c^	2.23 ± 0.28 ^a^	7.81 ± 0.63 ^a^
	500	3.63 ± 0.20 ^a^	8.30 ± 0.28 ^a^	1.45 ± 0.09 ^b,c^	3.37 ± 0.13 ^c^	2.32 ± 0.09 ^a^	4.49 ± 0.13 ^b,c^	2.04 ± 0.14 ^a^	8.51 ± 0.37 ^a^
	2500	2.82 ± 0.21 ^b^	7.02 ± 0.43 ^b^	1.26 ± 0.06 ^c^	2.89 ± 0.10 ^d^	2.48 ± 0.09 ^a^	4.71 ± 0.13 ^a,b^	2.07 ± 0.16 ^a^	7.98 ± 0.27 ^a^
**Mixture**	0	2.86 ± 0.14 ^b^	6.02 ± 0.33 ^b^	1.94 ± 0.10 ^a^	4.23 ± 0.14 ^a^	2.21 ± 0.11 ^a,b^	4.31 ± 0.09 ^a^	1.69 ± 0.10 ^b^	8.95 ± 0.26 ^a^
	20	3.77 ± 0.18 ^a^	7.43 ± 0.18 ^a^	1.27 ± 0.11 ^c,d^	3.68 ± 0.14 ^b^	2.51 ± 0.13 ^a^	3.86 ± 0.11 ^b^	1.93 ± 0.10 ^b^	6.83 ± 0.17 ^b^
	100	3.08 ± 0.14 ^b^	7.89 ± 0.31 ^a^	1.43 ± 0.07 ^b,c^	3.11 ± 0.11 ^c^	2.06 ± 0.11 ^b^	3.86 ± 0.11 ^b^	2.39 ± 0.11 ^a^	6.50 ± 0.18 ^b^
	500	2.97 ± 0.11 ^b^	7.91 ± 0.26 ^a^	1.62 ± 0.09 ^b^	3.76 ± 0.10 ^b^	2.06 ± 0.11 ^b^	3.79 ± 0.13 ^b^	2.30 ± 0.11 ^a^	6.69 ± 0.17 ^b^
	**2500**	2.87 ± 0.14 ^b^	7.70 ± 0.18 ^a^	** 1.05 ±0.09 ^d^**	3.24 ± 0.15 ^c^	**1.69 ± 0.08 ^c^**	3.75 ± 0.07 ^b^	1.84 ± 0.13 ^b^	6.71 ± 0.16 ^b^

Each value is the mean of three replicates ± SE (*n* = 30). Means with different letters (a, b, c, etc.) indicate significant differences at *p* < 0.05 level according to Fisher’s LSD test.Bolded values represent the most potent effects of the alkaloids with the most significant phytotoxic activity.

**Table 3 toxins-13-00706-t003:** IC_50_ (mg/mL) values of selected *S. alopecuroides* alkaloids on receiver species via Petri dish assays.

Alkaloids	*A. retroflexus*		*M. sativa*		*L. perenne*		*S. viridis*		Average Value of IC_50_			
	Root	Shoot	Root	Shoot	Root	Shoot	Root	Shoot	Root	Shoot	Root + Shoot	
**(1) Aloperine**	0.171	0.221	0.184	0.452	1.739	1.813	0.779	2.203	**0.718**	**1.172**	**0.945**	
(2) Matrine	0.674	1.694	1.141	2.113	1.813	3.298	2.117	2.704	1.436	2.452	1.944	
(3) Oxymatrine	0.525	1.343	0.232	0.603	4.695	3.805	3.907	11.246	2.340	4.249	3.295	
(4) Oxysophocarpine	0.259	0.620	0.482	/	4.905	/	3.026	/	2.168	0.620	1.394	
**(5) Sophocarpine**	0.475	0.464	0.351	0.155	0.474	1.194	0.958	1.459	**0.565**	**0.818**	**0.692**	
(6) Sophoridine	0.422	0.905	1.039	2.268	2.422	4.540	0.699	3.172	1.146	2.721	1.934	
(7) Average of individual alkaloid	0.421	0.875	0.572	1.118	2.675	2.930	1.914	4.057	1.396	2.005	1.701	
**(8) Mixture**	0.178	0.791	0.275	0.457	2.038	3.409	1.786	2.675	**1.069**	**1.833**	**1.451**	

IC_50_: the inhibitory concentration required for 50% inhibition. /: represents an invalid value. Bolded values represent the most potent effects of the alkaloids with the most significant phytotoxic activity.

**Table 4 toxins-13-00706-t004:** IC_50_ (mg/mL) values of selected *S. alopecuroides* alkaloids on receiver species via pot experiments.

Alkaloids	*A. retroflexus*		*M. sativa*		*L. perenne*		*S. viridis*		Average Value of IC_50_		
	Root	Shoot	Root	Shoot	Root	Shoot	Root	Shoot	Root	Shoot	Root + Shoot
(1) Aloperine	15.493	11.730	3.634	4.139	12.905	6.350	31.652	10.911	15.921	8.283	12.102
**(2) Matrine**	2.756	3.645	3.323	16.524	8.810	8.324	3.616	9.683	**4.626**	**9.544**	**7.085**
(3) Oxymatrine	1.550	2.382	4.203	7.428	12.065	4.905	12.086	3.587	7.476	4.576	6.026
(4) Oxysophocarpine	1.711	3.027	4.381	11.219	11.401	/	15.896	43.086	8.347	19.111	27.458
**(5) Sophocarpine**	3.931	3.646	9.602	4.177	3.315	7.148	9.247	8.552	**6.524**	**5.881**	**6.203**
(6) Sophoridine	5.035	5.188	/	472.484	3.264	6.190	37.014	6.668	15.104	122.634	68.869
(7)Average of individual alkaloid	5.079	4.936	5.029	85.995	8.627	6.583	18.252	13.748	9.666	28.338	21.291
**(8) Mixture**	2.571	3.422	4.857	40.489	9.108	5.190	3.411	/	**4.987**	**16.637**	**10.677**

IC _50_: the inhibitory concentration required for 50% inhibition. /: represents an invalid value. Bolded values represent the most potent effects of the alkaloids with the most significant phytotoxic activity.

## Data Availability

Not applicable.

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
