# Peer review of "Phytotoxic Activity of Alkaloids in the Desert Plant Sophora alopecuroides"

_toxins, 2021, doi:10.3390/toxins13100706_

Round 1

Reviewer 1 Report

This paper reports a study on the allelopathic activity of Sophora alopecturoides plant, investigating the phytotoxicity of the ethanol extract and total alkaloids obtained from its seeds. The phytotoxic effect of the main alkaloids produced by the plant tested one by one and as mixture has been also investigated.

The manuscript is overall well written and the results are adequately described. The extensive phytotoxicity investigation carried out both in vitro and via Pot experiments is of interest to the Toxins journal readership. I therefore suggest publication of the paper in Toxins after some minor points have been corrected.

Revisions to be addressed:

The authors name and affiliations are missing on the title page.

In Figure 2 some corrections should be done: 1) The chemical structures of Matrine and Sophoridine are exchanged. Consequently, also structure of Oxymatrine is wrong. 2) Hydrogens not present on stereogenic centres (i.e. with neither wedged nor dashed bonds) should be omitted on chemical structures. 3) In the structure of Aloperine hydrogens on the two stereogenic centres must be shown.

The caption of Figure 3 in incomplete and not clear. It should be specified that the phytohormone content of M. sativa is reported upon addition of the alkaloids mixture in different concentrations.

In the Materials and Methods section (page 17, line 358) ammonia concentration should be reported.

Typos: page 17 line 354: “sonication”

Reviewer 2 Report

The manuscript “Phytotoxic Activity of Alkaloids in the Desert Plant Sophora alopecuroides” submitted to Toxins describes allelopathic activity of alkaloids produced by S. alopecuroide on various receivers plants. The scope of the research is certainly of interest for the readers of the journal. The manuscript is generally well written and would benefit from editing for clarification of specific points and protocols. Detailed comments below.

Abstract: abbreviations should be avoided, line 7: ‘petri dish assay and pot experiment’ - explain what these experiments are.

Line 8 - what are the 4 receiver species? 

Introduction: 

Well structured, basic the overview of the research topic, however, I suggest to include any information about the receiver plant. The novelty of the research should also be highlighted.

Line 98, 201 (Figure 1 and 3) - The authors indicated that each value is the mean of three replicates ± SE and in parenthesis there is n = 30. What does ’n’ stand for? Are there three replicates or 30?

Lines 101-104 and Figure 2 - I suggest to move these elements to introduction. Chemical structures of alkaloids are not the results of the study and Figure shouldn’t be included in the Results section.

Line 105 -  Are the interactions between alkaloids  considered in the mixture experiments?  

Table 1 and 2 - tables contain a lot of data. I suggest to bold/color/highlight the most important results to make these tables easier to read. 

Figure 3 - Explain the abbreviations used in the graph. What is ‘w’ in Y axis? Figures and the captions should be self-explanatory. 

Line 354 - sonitation? Shouldn’t it be ’sonication’?

Reviewer 3 Report

The article is written in good scientific English with all the needed scientific attributes. It has a novelty, the content is clearly presented to readers, the data are presented at a high scientific level. Besides that, there are few minor points that should be corrected: Fig. 3 - please write the X-axis title, Table 2 and 3 - if it is possible please mark by bold the the most significant differences in the data series for each of the parameters. Fig 4 - make the X-axis signature 1 for all 4 figures inside, it's repeating 4 times in 1 figure, why? Please, little widen the conclusions. It should be proved by the data - now there is no any data proved the conclusion sentenses.

Round 2

Reviewer 2 Report

Authors responded to my comments and improved manuscript. 

Author Response

All authors greatly appreciate the reviewer's efforts in improving the quality of our manuscript.